# High Dietary Organic Iron Supplementation Decreases Growth Performance and Induces Oxidative Stress in Broilers

**DOI:** 10.3390/ani12131604

**Published:** 2022-06-21

**Authors:** Miaomiao Han, Xinsen Fu, Xiangqi Xin, Yuanyang Dong, Zhiqiang Miao, Jianhui Li

**Affiliations:** College of Animal Science, Shanxi Agricultural University, Taigu 030801, China; h_miaomiao2019@163.com (M.H.); fuxinsen2022@163.com (X.F.); xinxiangqi1998@163.com (X.X.); yuanyangdongemail@126.com (Y.D.); mzhq1981@163.com (Z.M.)

**Keywords:** high dietary iron, biochemical parameters, antioxidant status, iron transporter genes, broilers

## Abstract

**Simple Summary:**

Iron (Fe) is an essential trace element nutrient that plays a vital role in metabolic processes, such as oxygen transportation and electron transfer during respiration. However, exposure to high levels of dietary Fe can have negative effects on growth performance and antioxidant function in broilers. Organic Fe has a considerably higher bioavailability than inorganic Fe. To evaluate the safety of Fe chelates with lysine and glutamic acid (Fe–LG) in broilers, we investigated the effects of high concentrations of dietary Fe–LG on serum biochemical parameters, antioxidant status, and duodenal Fe transporter mRNA expression in broilers. A supplementation of 800 mg/kg Fe induced kidney function injury and liver oxidative stress in broilers, and decreased duodenal Fe transporter mRNA expression. Hence, this study demonstrated that high levels of Fe–LG negatively affect broiler health, and proposed a maximum supplemental dose of Fe–LG in broiler diet as a nutritional feed additive.

**Abstract:**

Although Iron (Fe) is an essential nutrient that plays a vital role in respiratory processes, excessive Fe in the diet can affect the health of broilers. We investigated the effects of diet supplemented with high levels of iron chelates with lysine and glutamic acid (Fe–LG) on the growth performance, serum biochemical parameters, antioxidant status, and duodenal mRNA expression of Fe transporters in broilers. A total of 800 1-day-old male Arbor Acres broilers were assigned to 5 groups, with 8 replicates each. Broilers were fed a corn–soybean meal basal diet or basal diets supplemented with 40, 80, 400, or 800 mg Fe/kg as Fe–LG for 6 weeks. The body weight (BW) was increased in the 80 mg Fe/kg treatment group, but decreased in the 800 mg Fe/kg treatment group on day 21. During days 1–21, compared with the control group, the supplementation of the 80 mg Fe/kg increased the average daily gain (ADG) and average daily feed intake (ADFI); however, the supplementation of the 800 mg Fe/kg group decreased the ADG and increased the FCR in broilers (*p* < 0.05). The heart, liver, spleen, and kidney indices were reduced in the 800 mg Fe/kg treatment group (*p* < 0.05). The supplementation of the 800 mg Fe/kg group increased the serum aspartate aminotransferase activity and the levels of creatinine and urea nitrogen on day 42 (*p* < 0.05). The broilers had considerably low liver total superoxide dismutase activity and total antioxidant capacity in the 800 mg Fe/kg treatment group (*p* < 0.05). Serum and liver Fe concentrations were elevated in the 400 and 800 mg Fe/kg treatment groups, but were not affected in the 40 and 80 mg Fe/kg treatment groups. The duodenal Fe transporters divalent metal transporter 1 (DMT1) and ferroportin 1 (FPN1) were downregulated in the Fe–LG treatment groups (*p* < 0.05). We conclude that a high dietary supplement of 800 mg Fe/kg in broilers leads to detrimental health effects, causing kidney function injury and liver oxidative stress.

## 1. Introduction

Iron (Fe) is a trace element that serves as an essential nutrient for animals, and plays an important role in oxygen transportation, the synthesis of nucleic acids, and electron transfer during respiration [1]. In addition, Fe is a vital cofactor of many enzymes, such as acetyl coenzyme A, succinodehydrogenase, xanthine oxidase, and cytochrome reductase [2]. Iron is transported to small intestinal epithelial cells via divalent metal transporter 1 (DMT1), after the ferric form (Fe^3+^) is reduced by duodenal cytochrome b to the ferrous form (Fe^2+^) [3]. Subsequently, Fe is stored in small intestinal epithelial cells until it is transported to the bloodstream via ferroportin (FPN1) [4]. Studies show that diets supplemented with appropriate doses of organic Fe can improve growth performance, elevate immune and antioxidant status, and enhance egg quality [5,6]. The National Research Council states that the Fe requirement in broilers is approximately 80 mg/kg [7]. However, research on poultry nutrition has shown that the recommendations vary widely from 45–136 mg Fe/kg [8]. An Fe requirement of 97–136 mg Fe/kg dry matter (DM) in a broiler diet has been reported to enhance the expression of Fe-containing enzymes in the liver or heart [9,10]. However, when Fe intake exceeds the nutritional requirement, excess Fe deposits in the liver. Fe overload can induce oxidative stress [11], impair immunity by releasing inflammatory cytokines, accelerate hepatocyte apoptosis, and cause serious structural and functional damages to the liver, heart, or intestines [12,13]. The administration of high dietary Fe content (500 mg Fe/kg DM) in the form of Fe sulfate heptahydrate reduces abdominal adipose fat deposition and liver triglyceride accumulation in male Ross 308 broilers [14]. In addition, Fe overload may lead to the pathogenesis of atherosclerosis, since excess Fe is closely related to reactive oxygen species (ROS) generation, lipoprotein oxidation, and platelet activation in mice [15,16]. Therefore, the potential toxic effects of high dietary Fe should be considered in broilers.

Fe, as a nutritional feed additive, is primarily present in inorganic forms as ferrous sulfate, in organic acid forms as fumarate and citrate, or as amino acid chelates with glycine and methionine [4,17]. Amino acid chelates of Fe have considerably higher bioavailability and stability than inorganic forms [6]. Furthermore, organic forms are environmentally safe and less toxic than inorganic forms. Fe chelates with lysine and glutamic acid (Fe–LG) represent a new form of complex amino acid chelates, which is regarded as a substitute for other authorized Fe additives, and does not increase the environmental burden of Fe [18]. Fe–LG has several nutritional functions derived from Fe, lysine, and glutamic acid. Lysine is important for muscle and immunity development in chickens [19,20]. Glutamic acid is necessary for the biosynthesis of glutamine, a key neurotransmitter [21]. Diet supplemented with 590 Fe mg/kg in the form of Fe–LG does not affect the feed conversion ratio (FCR) in 36-day-old Ross 308 male chickens [18]. To date, the effects of high dietary Fe administration in the form of Fe–LG on biochemical parameters and Fe transporter mRNA expression in broilers have not been investigated extensively. Therefore, the objective of this study was to investigate the effects of high dietary Fe administration in the form of Fe–LG on growth performance, organ index, serum biochemical parameters, antioxidant status, and Fe transporter mRNA expression in broilers, as well as to determine the maximum supplemental dose of Fe–LG in broiler diets as a nutritional feed additive.

## 2. Materials and Methods

### 2.1. Experimental Material

Iron chelates with lysine and glutamic acid (Fe-LG) were provided by Zinpro Animal Nutrition Technology Co., Ltd. (Shanghai, China). The active substrate is divalent Fe in the form of chelates of lysine and glutamic acid in a 1:1 mixture The structure formulas are presented in Figure 1. The chemical formulas of the two compounds are C_6_H_17_ClFeN_2_O_7_S (a) and C_5_H_12_FeNNaO_10_S (b), respectively [18]. The analyzed content of Fe was 154.2 g/kg.

### 2.2. Animals, Diets, and Experimental Design

A total of 800 1-day-old male AA broilers (average initial body weight, 42.46 ± 0.43 g) were purchased from Arbor Acres Poultry Breeding Company (Beijing, China), and were randomly divided into 5 groups, with 8 replicates per group, and 20 broilers per replicate. Broilers were fed the basal diet, or the basal diet supplemented with 40, 80, 400, and 800 mg Fe/kg in the form of Fe–LG, respectively. The Fe–LG were premixed into corn flour and then added to experimental diets. The diets were fed in pelleted form, and the experiment lasted for 6 weeks. The broilers were fed the starter diets from day 1 to 21, and broilers were fed the grower diets from day 22 to 42. All nutrients contained in the experimental diets met or exceeded the requirements based on the Arbor Acres broiler guide [22]. The ingredients and chemical compositions of the basal diets are shown in Table 1.

All broilers were raised in stainless steel cages in an environmentally controlled room. The initial temperature was maintained at 34 °C, which was gradually reduced until it reached at 21 °C, and was then maintained at this temperature until the end of the experiment. The relative humidity of the rearing room was kept between 50–70%. A total of 23 hours of light was maintained at first week, then 20 hours illumination per day for 2–4 weeks, and, finally, 23 hours light per day until the end of the experiment. Feed and water were provided ad libitum throughout the study.

### 2.3. Growth Performance and Sample Collection

On day 21 and 42, the body weight (BW) and feed consumption of all chickens in each replicate were recorded after 12 h fasting. The average daily gain (ADG), average daily feed intake (ADFI), and the ratio of feed to gain (FCR) were calculated on a replicate basis in the periods between days 1–21, 22–42, and 1–42. 

Then, 1 broiler in each replicate close to the average BW was selected for blood and tissue sample collection on day 21 and 42. Blood was collected via a wing vein using 0.8-mm-diameter needles into 10-mL anticoagulant-free vacutainer tubes, and was then clotted at room temperature for 30 min and centrifuged at 1500× *g* for 10 min in a 4 °C centrifuge. Serum was stored at −20 °C for later analysis of serum biochemical parameters of broilers. 

On day 42, after blood sampling, broilers were euthanized by jugular vein bleeding after stunning using 60% concentration of CO_2_ gas. The heart, liver, spleen, lung, kidney, thymus, and bursa of Fabricius were collected and weighted; then, sections of the liver were harvested and stored at −80 °C to analyze the antioxidant parameters and Fe content in the liver. The duodenum tissue samples were collected and immediately flash-frozen in liquid nitrogen for later analysis of Fe transporter mRNA expression.

### 2.4. Diet Fe Contents Analysis 

The iron concentrations in diets were determined by atomic absorption spectrometry (SavantAA2, GBC Scientific Equipment, Inc, Braeside, Australia), according to the reported method [23].

### 2.5. Organ Index

The organ indices of the heart, liver, spleen, lung, kidney, thymus, and bursa of Fabricius were calculated via the formula: organ index (%) = organ weight (g)/body weight (g) × 100%.

### 2.6. Analysis of Serum Biochemical Parameters

On day 21 and 42, the contents of alkaline phosphatase (ALP), alanine aminotransferase (ALT), aspartate aminotransferase (AST), albumin (ALB), total protein (TP), creatinine (CREA), urea nitrogen (UN), glucose (GLU), total antioxidant capacity (T-AOC), catalase (CAT), and malondialdehyde (MDA) in serum were measured by the corresponding commercial kits purchased from Nanjing Jiancheng Bioengineering institute (Nanjing, China), according to the manufacturer’s instructions. Serum Fe contents were detected according to the method reported by Wu et al [24]. The serum biochemical parameters with the corresponding commercial kit number are listed in Table 2.

### 2.7. Analysis of Antioxidant Status in Liver

The activities of total superoxide dismutase (T-SOD), T-AOC, and MDA concentrations in the livers of 42-day-old broilers were detected by the corresponding commercial kits purchased from Nanjing Jiancheng Bioengineering institute (Nanjing, China), according to the manufacturer’s instructions. Liver Fe contents were detected according to the method reported by Yuan et al [25]. The serum biochemical parameters with the corresponding commercial kit number are listed in Table 3.

### 2.8. Quantitative Real-Time PCR (qPCR)

Total RNA was extracted from the duodenum using TRIzol reagent (catalog number 9108, Takara, Dalian, China). The concentration and purity of the total RNA were evaluated by the ratio of 260 and 280 nm using a spectrophotometer (P330-31, Implen Nano Photometer, München, Germany). The reverse transcriptions of mRNA were conducted using the PrimeScriptTM RT reagent kit with gDNA Eraser (catalog number RR047A, Takara Bio Inc., Otsu, Japan) according to the manufacturer’s instructions. Quantitative real-time PCR was performed in triplicate on the Step One Plus Real-Time PCR system, using the SYBR○R Premix Ex TaqTM II (Tli RNaseH Plus) (catalog number RR802A, Takara Bio Inc., Otsu, Japan) according to the manufacturer’s instructions. The following procedure was used in the PCR program: 1 cycle at 95 °C for 30 s, then 95 °C for 5 s, and 60 °C for 30 s, then 45 cycles at 95 °C for 5 s, and 60 °C for 30 s. The primer information of DMT1, FPN1, and β-actin (housekeeping gene) is summarized in Table 4. All gene sequences were synthesized by Sangon Biotech (Shanghai) Co., Ltd. The relative abundance of mRNA was calculated using the 2^−ΔΔCt^ method, with the quantity of the control group scaled to 1. The geometric mean of β-actin was selected as an internal reference gene to standardize the expression of target genes. 

### 2.9. Statistical Analysis

All of the data were analyzed by one-way ANOVA for a completely randomized design, using the general linear model (GLM) procedure of SAS [29]. The normality in data distribution was evaluated by the Shapiro–Wilk test. The equality of variance among groups were analyzed by the Chi-square test. The differences among treatments were determined with the least-significance-difference (LSD) test. For the growth performance, the replicate unit was a pen with 20 birds, and for all other parameters, an individual bird was considered as the experiment unit. A probability level of *p* < 0.05 was considered statistically significant. 

## 3. Results

### 3.1. Fe Content of the Diets

The Fe contents of the experimental diets are listed in Table 5. The Fe contents of the control diets were 105.77 and 102.85 mg/kg in the starter and growth phase, respectively. The Fe concentrations of the experimental diets supplemented with different levels of Fe–LG were in agreement with the expected values. 

### 3.2. Growth Performance and Organ Index

The effects of dietary supplementation with high doses of Fe–LG on broiler growth performance are shown in Table 6. On day 21, there was a significant increase in the BW of the group fed with 80 mg Fe/kg; however, a growth reduction with decreased BW was observed in the group fed with 800 mg Fe/kg compared, with that in the control group (*p* < 0.001). The ADG and ADFI were elevated in broilers fed with 80 mg Fe/kg; however, the supplementation of 800 mg Fe/kg significantly decreased ADG and increased FCR on days 1–21 (*p* < 0.05). Diets supplemented with 80 and 400 mg Fe/kg significantly decreased the ADG on days 21–42 (*p* < 0.05), compared with that in the control group. There were no significant differences in ADG, ADFI, or FCR in broilers among the groups during 1–42 days. 

The broilers fed with diet containing 80 mg Fe/kg had a higher liver index than that of broilers fed with 0, 400, or 800 mg Fe/kg (*p* < 0.001; Table 7). The organ indices of the heart, liver, spleen, and kidneys were significantly reduced in the groups treated with the highest Fe dose (*p* < 0.01). The index of the bursa of Fabricius was lower in birds that received a diet with 400 mg Fe/kg, compared with that of broilers in the remaining groups (*p* < 0.05). There were no differences in the lungs or thymus indices among the groups.

### 3.3. Effects of Fe–LG Administration on Blood Parameters 

The effects of dietary supplementation with high doses of Fe–LG on the serum biochemical parameters of broilers are presented in Table 8. No differences were observed in the activities of serum AST, ALB, and T-AOC, nor the contents of TP, CREA, UN, and GLU among all treatments in broilers on day 21. However, the supplementation of 800 mg Fe/kg significantly decreased ALP activity, compared to that in the control group (*p* < 0.001). CAT activity was elevated in broilers fed diets containing 40 and 80 mg Fe/kg (*p* < 0.001). Broilers fed diets containing 400 and 800 mg Fe/kg had considerably higher Fe and MDA concentrations, but lower CAT activity in the serum, than those in other groups (*p* < 0.01). 

On day 42, the activity of ALT and the concentrations of TP, GLU, and MDA were not affected by dietary supplementation with different levels of Fe. Higher AST activities and Fe contents, but lower T-AOC capacities, were found in the 400 and 800 mg Fe/kg treatment groups than those in the control group (*p* < 0.05). In addition, supplementation of 800 mg Fe/kg elevated the serum CREA and UN levels (*p* < 0.05). Moreover, broilers fed with a 400 mg Fe/kg diet had higher ALB content and lower CAT activity than those in the control group (*p* < 0.01).

### 3.4. Effects of Fe–LG Administration on the Oxidative Status of the Liver

Supplementation with 400 and 800 mg Fe/kg significantly decreased the T-SOD activity and increased Fe levels in the liver, as compared with those in the control group (*p* < 0.01; Table 9). However, supplementation of 40 and 80 mg Fe/kg had no significant effect on T-AOC and T-SOD activities, nor MDA and Fe concentrations in the liver. 

### 3.5. Effects of Fe–LG Administration on DMT1 and FPN1 mRNA Expression in the Duodenum

Supplementation with different levels of Fe significantly downregulated DMT1 and FPN1 in the duodenum of 42-day-old broilers compared with the levels in the control group (*p* < 0.05; Figure 2). An increase in the dose of supplemental Fe contributed to a reduction in the mRNA expression of the two transporters, with significant differences between groups which were administered the higher doses (400 and 800 mg Fe/kg) and the lower dose (40 mg Fe/kg) (*p* < 0.05), respectively. 

## 4. Discussion

Fe is an essential trace mineral, and is vital for the growth performance of poultry [30]. However, Fe overload can lead to oxidative stress, which disrupts cellular redox balance [11]. In the current study, diets supplemented with 40 mg Fe/kg did not have any effect on growth performance in AA broilers, which is consistent with the result reported by the European Food Safety Authority (EFSA) of 40 mg Fe/kg in the form of Fe–LG in Ross 308 broilers [18]. Lin et al. [8] indicated that the growth performance in Lingnan yellow broilers was not influenced by diets supplemented with 50–150 mg/kg in the form of FeSO_4_·7H_2_O. However, the present study indicated that diets supplemented with 80 mg Fe/kg increased BW, ADG, and ADFI in 21 days during the starter phase. Our previous study has investigated the recommended supplemental level of Fe–LG (40–80 mg/kg) in AA broilers [31]. Ma et al. [9] indicated that diets supplemented with 40–60 mg Fe/kg in the form of FeSO_4_·7H_2_O increased the BW, but those supplemented with 80–120 mg Fe/kg did not experience growth performance effects from day 1 to 21. In addition, previous studies indicated an optimal Fe dose of 127 and 118 mg/kg in the form of FeSO_4_·7H_2_O, based on maximal growth performance parameters [9,32]. Since growth performance is always influenced by animal breeds, diet type, etc., it is usually not a sensitive indicator for evaluating the Fe requirements for broilers.

According to the Guidance on the assessment of the safety of feed additive [33], the Fe use level was set at 80 mg/kg, according to our previous study [31], and the levels of overdose groups were set at 400 mg/kg (5× use-level) and 800 mg/kg (10× use-level) in the current study. Discrepancies have been found in previous findings on the effect of high dietary Fe administration on the growth performance of broilers. In the current study, the supplementation with a diet containing 800 mg Fe/kg decreased the BW of broilers on day 21 and ADG on days 1–21, and it elevated the FCR in the starter phase. Therefore, the supplementation of 800 mg Fe/kg had adverse effects on the growth performance of AA broilers in the starter phase. In addition, there were no differences in ADFI and FCR during the growth phase, nor during the entire experimental period, which may be due to reduced Fe absorption with advancing age [34]. The current result was consistent with previous findings, which reported that high dietary Fe administration of 690 mg Fe/kg in the form of ferrous sulfate decreases the growth performance of broilers during days 1–21 [14]. However, the EFSA [18] reported that BW, feed intake, and FCR in Ross 308 chickens were not affected by diets containing 290 or 590 mg Fe/kg in the form of Fe–LG, whereas the BW and feed intake are increased by supplementation with 590 mg Fe/kg in the form of ferrous sulfate. However, no obvious effects were found in FCR, regardless of supplementation levels (290 or 590 mg Fe/kg) or forms (Fe–LG or Fe sulfate). Gou et al. [30] reported that no effects on ADFI, ADG, or FCR were observed in Lingnan broilers fed with 700 and 1400 mg Fe/kg in the form of ferrous gluconate during days 1–21. Therefore, different effects on growth performance may be attributed to animal stages and breeds, Fe doses, and the bioavailability of Fe sources. The organ index is an important parameter reflecting the status of organ development in animals [35]. Bai et al. [14] reported that high dietary Fe supplementation (500 mg Fe/kg) in the form of Fe sulfate heptahydrate does not affect liver weight and liver yield during days 1–21 in Ross 308. In the current study, the liver indices on day 42 decreased in broilers fed diets supplemented with 400 and 800 mg Fe/kg. Furthermore, supplementation with 800 mg Fe/kg significantly decreased the heart, spleen, and kidney indices, which may be induced by the Fe deposition in bodies or the changes in lipid metabolism owing to high dietary Fe supplementation in broilers [14]. 

Biochemical parameter analyses represent valuable indicators for assessing the intensity and direction of metabolic changes that occur in organisms or their response to feed additives [36]. The liver is the main target organ for detoxification of toxins and heavy metals, other than the kidneys [37]. Serum enzymes, such as ALP, ALT, and AST, serve as biomarkers of liver function [38]. ALP is mainly produced in the liver, kidney, and bones, and it increases inorganic phosphate levels and activates collagen fibers in the bone matrix; thus, ALP contributes to bone mineralization, as well as the functions of the liver and kidney [39,40]. In the current study, serum ALP activity decreased in broilers supplemented with a high dose of Fe–LG, which was consistent with the results reported by Yamasaki et al. [41], who reported that high Fe exposure may inhibit ALP activity in osteoblastic cells. AST and ALT are cytosolic and mitochondrial enzymes. The leakage of these two enzymes into the bloodstream indicates damage to the cell membrane or mitochondrial membrane integrity, as well as permeability caused by the ingestion of harmful materials, such as toxins and metals, biliary cholestasis, and hyperplasia of bile ducts [42,43]. ALT is usually confined to liver cells, and is more specific than AST as a biomarker of liver damage, since AST is present in various organs, such as the kidneys, heart, muscles, brain, and lungs [44]. In the present study, ALT and AST activities were increased in broilers supplemented with 800 mg Fe/kg, indicating that high Fe exposure may cause broiler liver damage to a certain extent, which is in agreement with the results reported by Gholampour and Saki [45]. Total protein (TP) is vital for the maintenance of homoeostasis in broilers. ALB content accounts for the largest protein fraction (40–60%), and it is synthesized in the liver [46]. Both TP and ALB contents in the blood are considered markers of liver function that reflect protein synthesis [47]. TP and ALB levels decrease in rats under ferrous sulfate-induced oxidative stress conditions [45,48]. In the current study, TP content was not affected by Fe supplementation; however, serum ALB levels increased in broilers fed with 400 mg Fe/kg. Excess Fe intake leads to ROS generation in the bloodstream, and Fe^2+^ is then oxidized to insoluble Fe^3+^ [49]. Serum ALB plays an important role in the transport, metabolism, and distribution of water-insoluble components, such as fatty acids and metal ions. ALB can bind to various metal ions with different geometries and metal-coordination properties. The binding of Fe^3+^ to ALB minimizes the toxic effects of Fe overload. The final complex is soluble in water, and is excreted in urine [50]. Serum CREA and UN serve as endogenous indicators for estimating the glomerular filtration rate to reflect kidney state, and are released into plasma at a relatively constant rate under normal conditions [51]. In the present study, CREA and UN levels were elevated in broilers fed with 800 mg Fe/kg. This may be explained by the fact that high-dose Fe exposure damages kidney functions, and CREA and UN were mainly excreted via the glomerulus of the kidney, which is consistent with the results reported previously [45,48]. Hence, high dietary Fe levels may lead to liver and kidney function disorders in broilers. 

In the current study, supplementation with 40 and 80 mg Fe/kg did not affect the serum and liver Fe contents, which was consistent with a previous study involving diet supplemented with 50–150 mg Fe/kg (FeSO_4_·H_2_O) in Chinese yellow male broilers [8]. The Fe content in plasma was not influenced on day 14, but increased on day 21 in AA male broilers fed with 60 mg Fe/kg (FeSO_4_·7H_2_O) [52]. In addition, Fe content was increased in the serum and liver after supplementation with 400 and 800 mg Fe/kg. This result is consistent with those reported previously [53,54]. However, the EFSA [18] reported that there were no differences in Fe content in the breast muscle, skin fat, kidney, liver, and tibia of Ross 308 broilers fed with Fe–LG (40, 290, and 590 mg Fe/kg). The different results for Fe depositions in broilers may be due to the distinct absorption pathways of different forms of Fe. Excessive Fe levels induce ROS generation and cell death [55]. T-AOC, SOD, CAT, and MDA are considered as indicators of antioxidant function. T-AOC levels reflect the condition of antioxidant defense systems, such as antioxidant enzymes and antioxidant substances [56]. Furthermore, SOD synthesis under stress conditions can promote the dismutation of superoxide radicals to molecular oxygen and hydrogen peroxide, and CAT can transform hydrogen peroxide into water and oxygen as the first line of defense against the deleterious effects of free radicals and ROS [57]. MDA is one of the final products of polyunsaturated fatty acid peroxidation in cells, and serves as an indicator of alterations in membrane fluidity and fragility [58]. In the present study, serum MDA content increased in broilers on day 21; however, serum and liver MDA contents on day 42 were not affected by the high Fe diet. However, T-AOC capacity and SOD activity in the livers of 42-day-old broilers decreased in the 800 mg Fe/kg treatment group. Gou et al. [30] indicated that diet supplementation of 700 and 1400 mg Fe/kg in the form of ferrous gluconate increased MDA content in the jejunal mucosa of Chinese yellow broilers on day 21, whereas plasma MDA levels were not affected by dietary Fe exposure. Supplemental 97–136 mg/kg (FeSO_4_·7H_2_O) had positive effects on the expression of Fe-containing enzymes, such as succinate dehydrogenase, catalase, or cytochrome c oxidase, in AA broilers [9]. Therefore, an appropriate dose of Fe^2+^ is important for oxygen transport, energy metabolism, and iron–sulfur protein production in mitochondria [59], whereas the accumulation of Fe may directly generate excessive ROS, inducing oxidative injuries [55]. Therefore, in this study, exposure to 800 mg Fe/kg in the form of Fe–LG induced oxidative stress. 

Divalent metal transporter 1 (DMT1) is distributed in the brush border of duodenal villi and crypts, and it transports free Fe^2+^ from the stomach into intestinal enterocytes [3]. Fe is stored in the small intestinal epithelial cells or transported to the bloodstream via FPN1, which is the sole identified transporter that promotes Fe efflux from enterocytes into the bloodstream [4]. Thus, DMT1 and FPN1 are important transporters for maintaining the balance of Fe in the body. A high Fe diet can downregulate the expression of DMT1 and FPN1 in mice, pigs, and bull calves [60,61,62]. In domestic chicken (Ross 308 broilers or Gallus gallus), DMT1 and FPN1 expression is downregulated in the duodenum under high dietary Fe conditions [53,63]. Similarly, the current study showed that DMT1 and FPN1 expression was downregulated in broilers fed diets supplemented with 400 and 800 mg Fe/kg. This reduction in FPN1 mRNA expression could downregulate the Fe efflux from enterocytes into the bloodstream in order to avoid injuries mediated by Fe overload. These results also provide evidence that the gene expression of Fe-related transporters may be related to the dietary Fe levels or the transport forms. However, it is not clear whether the absorption way of organic Fe from the bloodstream into intestinal epithelial cells is in the form of Fe^2+^ or the entire chelates form via peptide transporter [3,52], which requires further investigation.

## 5. Conclusions

Diets supplemented with 80 mg Fe/kg could improve the growth performance in broilers, while 800 mg Fe/kg had adverse effects on the growth performance during the starter phase. The organ indices of the heart, liver, spleen, and kidneys were reduced in broilers supplemented with 800 mg Fe/kg. High Fe intake caused liver and kidney injuries, as indicated by certain serum biochemical parameters. Oxidative stress was detected in the 800 mg Fe/kg treatment group, based on the decrease in antioxidant activity and the elevated Fe concentrations in the serum and liver. The downregulation of DMT1 and FPN1 in the duodenum in the 400 and 800 mg Fe/kg treatment groups could, in turn, downregulate Fe efflux from enterocytes into the bloodstream, and avoid relatively greater stress injuries caused by Fe overload. Hence, an overdose of Fe supplementation, provided as 800 mg Fe/kg in the form of Fe–LG, will have detrimental effects on broilers. 

## Figures and Tables

**Figure 1 animals-12-01604-f001:**
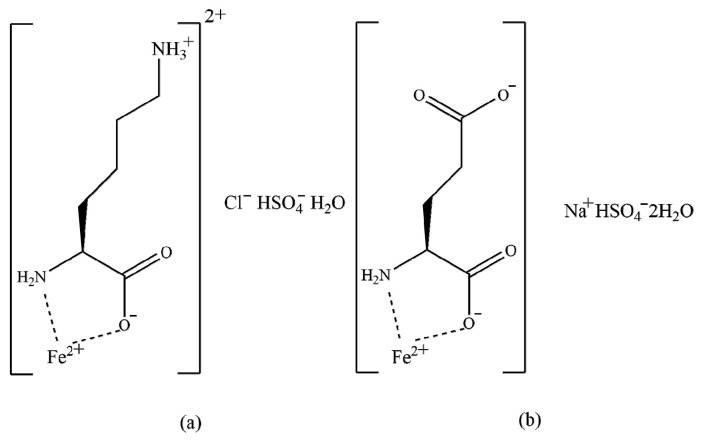
The structure formulas of iron chelates of lysine (**a**) and glutamic acid (**b**).

**Figure 2 animals-12-01604-f002:**
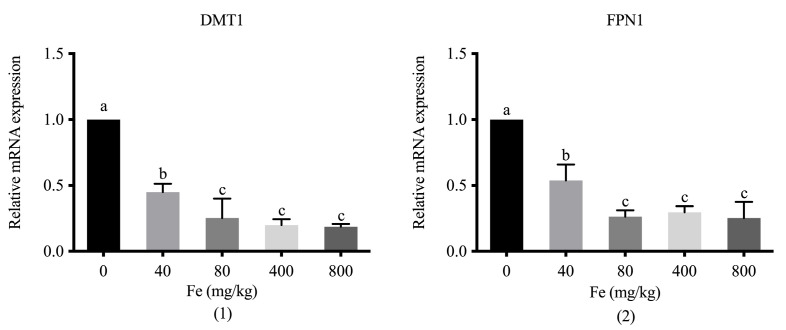
The relative mRNA expression of DMT1 (**1**) and FPN1 (**2**) in the duodenum of 42-day-old broilers fed with different doses of Fe–LG. DMT1, divalent metal transporter 1; FPN1, ferroportin 1. ^a–^^c^ Bars without the same letters differ significantly (*p* < 0.05).

**Table 1 animals-12-01604-t001:** Ingredient composition and nutrient level of the basal diets (air-dry basis).

Items	Starter Diet 1~21 Days of Age	Grower Diet 22~42 Days of Age
Ingredients (%)		
Corn	56.22	52.90
Wheat flour	3.10	7.00
Soybean oil	2.20	5.90
Soybean meal	33.26	30.16
Limestone	1.20	1.02
Dicalcium phosphate	1.74	1.30
Salt	0.30	0.30
L-Lysine	0.60	0.27
DL-Methionine	0.26	0.19
L-Threonine	0.20	0.06
L-Valine	0.12	0.06
Choline chloride	0.13	0.17
Phytase (5000 FTU/kg)	0.01	0.01
Complex phosphoesterasum ^1^	0.03	0.03
Vitamin premix ^2^	0.03	0.03
Mineral premix ^3^	0.60	0.60
Total	100.00	100.00
Nutrient composition ^4^		
ME (Kcal/kg)	2950	3200
CP (%)	20.93	19.47
Ca (%)	0.97	0.78
Available P (%)	0.47	0.38
Lys (%)	1.42	1.16
Met (%)	0.56	0.47
Met + Cys (%)	0.90	0.80
Fe (mg/kg)	105.77	102.85

^1^ The compound enzyme contains xylanase ≥7500 viscosity units/g and β-glucanase ≥170 AGL units/g. ^2^ Each kilogram of the diet provided in the multivitamin premix: Vitamin A, 9500 IU; Vitamin D_3_, 62.5 μg; Vitamin E, 30 IU; Vitamin K_3_, 2.65 mg; Vitamin B_1_, 2 mg; Vitamin B_2_, 6 mg; Vitamin B_12_, 0.025 mg; Biotin, 0.0325 mg; Folic acid, 1.25 mg; Pantothenic acid, 12 mg; Niacin, 50 mg. ^3^ Per kilogram of diet provided in the compound trace element premix: copper, 8 mg (CuSO_4_·5H_2_O); zinc, 90 mg (ZnSO_4_·H_2_O); manganese, 80 mg (MnSO_4_·H_2_O); iodine, 0.35 mg (Ca(IO_3_)_2_); Selenium, 0.3 mg (Na_2_SeO_3_). ^4^ All nutrient levels were analyzed except metabolizable energy. ME, metabolizable energy; CP, crude protein; Ca, calcium; P, phosphors; Lys, lysine; Met, methionine; Cys, cysteine; Fe, iron.

**Table 2 animals-12-01604-t002:** The serum biochemical parameters with the corresponding commercial kit number.

Serum Biochemical Parameter	Kit No.
Alkaline phosphatase (ALP)	A059-1-1
Alanine aminotransferase (ALT)	C009-2-1
Aspartate aminotransferase (AST)	C010-1-1
Albumin (ALB)	A028-2-1
Total protein (TP)	A045-2-2
Creatinine (CREA)	C011-2-1
Urea nitrogen (UN)	C013-2-1
Glucose (GLU)	F006-1-1
Total antioxidant capacity (T-AOC)	A015-1-2
Catalase (CAT)	A007-1-1
Malondialdehyde (MDA)	A003-1-2
Serum iron (Fe)	A039-1-1

**Table 3 animals-12-01604-t003:** The liver antioxidant index with the corresponding commercial kit number.

Liver Antioxidant Index	Kit No.
Total superoxide dismutase (T-SOD)	A001-1-1
Total antioxidant capacity (T-AOC)	A015-1-2
Malondialdehyde (MDA)	A003-1-2
Liver iron (Fe)	A039-2-1

**Table 4 animals-12-01604-t004:** Primer sequences for real-time fluorescent quantitative PCR ^1^.

Genes	Primer Sequences ( 5′—3′)	Product Length (bp)	Accession No.	References
DMT1	F: AGCCGTTCACCACTTATTTCG	129	GI 206597489	[26]
	R: GGTCCAAATAGGCGATGCTC			
FPN1	F: GAGACTGGGTGGACAAGAACTC	68	GI 61098365	[27]
	R: ATGCATTCTGAACAACCAAGGA			
β-actin	F: ACCTGAGCGCAAGTACTCTGTCT	95	NM_205518	[28]
	R: CATCGTACTCCTGCTTGCTGAT			

^1^ DMT1, divalent metal transporter 1; FPN1, ferroportin 1.

**Table 5 animals-12-01604-t005:** Iron contents in all experimental diets (mg/kg) ^1^.

Items	Fe Level in the Form of Fe–LG (mg/kg)				
	0	40	80	400	800
Day 1 to 21	105.77	149.51	175.92	503.12	907.23
Day 22 to 42	102.85	147.43	177.64	505.62	905.20

^1^ Each diet was analyzed in triplicate. Fe–LG, iron chelates with lysine and glutamic acid.

**Table 6 animals-12-01604-t006:** Effects of dietary supplementation with high doses of Fe–LG on growth performance of broilers ^1^.

Items	Fe Level in the Form of Fe–LG (mg/kg)					SEM	*p*-Value
	0	40	80	400	800		
BW (g)							
Day 1	42.37	42.25	42.34	42.69	42.65	0.144	0.271
Day 21	685.30 ^bc^	667.04 ^cd^	718.71 ^a^	702.03 ^ab^	651.63 ^d^	9.404	<0.001
Day 42	2565.82	2524.15	2513.74	2435.81	2530.87	30.184	0.092
Day 1–21							
ADG (g/d)	30.62 ^bc^	29.75 ^cd^	32.21 ^a^	31.40 ^ab^	29.00 ^d^	0.449	<0.001
ADFI (g/d)	46.89 ^bc^	46.53 ^bc^	48.75 ^a^	48.30 ^ab^	45.86 ^c^	0.571	0.006
FCR	1.53 ^bc^	1.56 ^ab^	1.51 ^c^	1.54 ^ab^	1.58 ^a^	0.015	0.015
Day 22–42							
ADG (g/d)	89.55 ^a^	88.43 ^ab^	85.48 ^bc^	82.56 ^c^	89.49 ^a^	1.420	0.009
ADFI (g/d)	138.13	133.80	135.25	132.47	137.21	2.174	0.506
FCR	1.54	1.51	1.59	1.61	1.53	0.023	0.099
Day 1–42							
ADG (g/d)	60.08	59.09	58.84	56.98	59.24	0.718	0.090
ADFI (g/d)	92.98	91.23	91.67	90.38	92.26	1.231	0.739
FCR	1.55	1.54	1.56	1.59	1.56	0.019	0.645

Data represent mean value of 8 replicates, with 20 birds each. ^a–d^ Mean values in the same row without the same superscripts differ significantly (*p* < 0.05). ^1^ Fe–LG, iron chelates with lysine and glutamic acid; BW, body weight; ADG, average daily gain; ADFI, average daily feed intake; FCR, the ratio of feed to gain; SEM, standard error of mean.

**Table 7 animals-12-01604-t007:** Effects of dietary supplementation with high doses of Fe–LG on organ indices of broilers on day 42 (%) ^1^.

Items	Fe Level in the Form of Fe–LG (mg/kg)					SEM	*p*-Value
	0	40	80	400	800		
Heart	0.32 ^a^	0.35 ^a^	0.36 ^a^	0.24 ^ab^	0.21 ^b^	0.042	0.002
Liver	1.64 ^b^	1.90 ^ab^	2.07 ^a^	1.27 ^c^	1.19 ^c^	0.140	<0.001
Spleen	0.09 ^a^	0.10 ^a^	0.11 ^a^	0.10 ^a^	0.05 ^b^	0.014	0.002
Lungs	0.43	0.42	0.45	0.34	0.37	0.039	0.087
Kidney	0.40 ^a^	0.47 ^a^	0.39 ^a^	0.36 ^ab^	0.26 ^b^	0.047	0.002
Thymus	0.26	0.24	0.26	0.20	0.21	0.037	0.262
Bursa of Fabricius	0.04 ^a^	0.04 ^a^	0.04 ^a^	0.02 ^b^	0.03 ^a^	0.004	0.012

Data represent mean values of eight birds in each treatment. ^a–c^ Mean values in the same row without the same superscripts differ significantly (*p* < 0.05). ^1^ Fe–LG, iron chelates with lysine and glutamic acid; SEM, standard error of mean.

**Table 8 animals-12-01604-t008:** Effects of dietary supplementation with high doses of Fe–LG on serum biochemical parameters of broilers ^1^.

Items	Fe Level in the Form of Fe–LG (mg/kg)					SEM	*p*-Value
	0	40	80	400	800		
Day 21							
ALP (U/L)	6053 ^a^	5760 ^a^	5769 ^a^	4233 ^ab^	2089 ^b^	519	<0.001
ALT (U/L)	38.05 ^ab^	36.44 ^b^	40.64 ^ab^	39.53 ^ab^	43.22 ^a^	0.746	0.034
AST (U/L)	256.10	273.50	249.10	294.60	311.30	8.604	0.121
ALB (g/L)	14.76	15.96	13.86	13.46	14.86	0.385	0.255
TP (g/L)	25.34	29.69	23.84	24.64	32.17	1.478	0.267
CREA (μmol/L)	41.63	35.93	44.30	46.98	36.63	1.857	0.426
UN (mmol/L)	3.92	3.75	4.07	3.88	3.60	0.123	0.518
GLU (mmol/L)	9.66	9.17	9.47	9.74	9.67	0.162	0.822
Fe (μmol/L)	24.03 ^c^	28.46 ^c^	29.27 ^c^	52.47 ^b^	56.70 ^a^	5.126	0.001
T-AOC (U/mL)	12.90	12.01	14.74	13.09	12.13	2.871	0.896
CAT (U/mL)	0.48 ^b^	0.66 ^a^	0.72 ^a^	0.31 ^c^	0.37 ^c^	0.059	0.001
MDA (nmol/mL)	1.83 ^c^	1.84 ^c^	2.74 ^b^	3.19 ^a^	3.06 ^a^	0.367	0.001
Day 42							
ALP (U/L)	1001 ^b^	1830 ^a^	1762 ^ab^	1300 ^ab^	944 ^b^	103	0.004
ALT (U/L)	22.60	23.08	24.38	24.94	23.90	0.474	0.532
AST (U/L)	258.40 ^b^	255.80 ^b^	262.40 ^b^	321.70 ^a^	329.50 ^a^	9.403	0.031
ALB (g/L)	10.60 ^b^	12.65 ^ab^	10.65 ^b^	13.60 ^a^	11.49 ^ab^	0.335	0.008
TP (g/L)	28.56	25.65	25.53	23.71	21.11	1.221	0.393
CREA (μmol/L)	41.44 ^b^	51.50 ^ab^	44.16 ^ab^	44.24 ^ab^	52.25 ^a^	1.295	0.014
UN (mmol/L)	0.74 ^b^	1.32 ^b^	1.45 ^b^	1.15 ^b^	1.96 ^a^	0.132	0.003
GLU (mmol/L)	11.45	11.10	11.51	11.23	12.08	0.190	0.545
Fe (μmol/L)	22.29 ^c^	25.07 ^c^	27.08 ^c^	37.68 ^b^	64.85 ^a^	2.251	0.001
T-AOC (U/mL)	13.79 ^a^	11.21 ^ab^	13.97 ^a^	10.37 ^b^	8.93 ^b^	1.388	0.007
CAT (U/mL)	0.55 ^ab^	0.67 ^a^	0.63 ^a^	0.39 ^c^	0.44 ^bc^	0.067	0.002
MDA (nmol/mL)	2.21	2.34	2.65	2.81	2.71	0.590	0.790

Data represent mean values of eight birds in each treatment. ^a–c^ Mean values in the same row without the same superscripts differ significantly (*p* < 0.05). ^1^ Fe–LG, iron chelates with lysine and glutamic acid; SEM, standard error of mean; ALP, alkaline phosphatase; ALT, alanine aminotransferase; AST, aspartate aminotransferase; ALB, albumin; TP, total protein; CREA, creatinine; UN, urea nitrogen; GLU, glucose; Fe, iron; T-AOC, total antioxidant capacity; CAT, catalase; MDA, malondialdehyde.

**Table 9 animals-12-01604-t009:** Effects of dietary supplementation with high doses of Fe–LG on antioxidant index and iron content in liver of 42-day-old broilers ^1^.

Items	Fe Level in the Form of Fe–LG (mg/kg)					SEM	*p*-Value
	0	40	80	400	800		
T-SOD (U mg/prot)	341.93 ^a^	337.94 ^ab^	358.75 ^a^	319.73 ^b^	334.92 ^b^	7.172	0.001
T-AOC (U mg/prot)	2.50 ^a^	2.92 ^a^	2.91 ^a^	2.32 ^ab^	1.47 ^b^	0.482	0.047
MDA (nmol mg/prot)	1.43	1.31	1.80	1.12	1.42	0.292	0.253
Fe (μg/g)	77.60 ^c^	87.46 ^c^	81.34 ^c^	104.60 ^b^	114.43 ^a^	4.091	0.001

Data represent mean values of eight birds in each treatment. ^a–c^ Mean values in the same row without the same superscripts differ significantly (*p* < 0.05). ^1^ Fe–LG, iron chelates with lysine and glutamic acid; SEM, standard error of mean; T-SOD, total superoxide dismutase; T-AOC, total antioxidant capacity; MDA, malondialdehyde.

## Data Availability

The data presented in this study are available on request from the corresponding author.

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
