# Peer review of "High Dietary Organic Iron Supplementation Decreases Growth Performance and Induces Oxidative Stress in Broilers"

_animals, 2022, doi:10.3390/ani12131604_

Round 1
Reviewer 1 Report
- There are too many grammar problems, therefore please ask native English professionals to polish and modify.
- Please discuss the criteria for setting supplemental levels, especially the two high supplemental levels.
- This conclusion doesn't make sense. According to the data, diet supplemented with 400 mg not 800 mgFe/kg had adverse effects on growth performance and organ development of broilers. The negative effects of high addition levels should be discussed and summarized according to different parameters.
Author Response
Response to Reviewer 1 Comments
Point 1: There are too many grammar problems, therefore please ask native English professionals to polish and modify.
Response 1: We have checked language and syntax. All changes are marked in blue font as highlighted amendments in the revised manuscript. We would like to thank the Editage (www.editage.cn) for scrutinizing and correcting the language.
Point 2: Please discuss the criteria for setting supplemental levels, especially the two high supplemental levels.
Response 2: According to the Guidance on the assessment of the safety of feed additive (ESFA, 2017), the design of a tolerance test should include a control group, a use-level group and an over dose group with a multifold of the use level. For the use-level, it was the recommendation level about 80 mg/kg of supplemental Fe. The levels of over dose groups were set at 5×use-level (400 mg/kg) and 10×use-level (800 mg/kg) in the current study.
Point 3: This conclusion doesn't make sense. According to the data, diet supplemented with 400 mg not 800 mgFe/kg had adverse effects on growth performance and organ development of broilers. The negative effects of high addition levels should be discussed and summarized according to different parameters.
Response 3: “In conclusion, diet supplemented with 800 mg Fe/kg in the form of Fe-LG had adverse effects on growth performance and organ development of broilers” had been changed into “Diet supplementation of 800 mg Fe/kg in the form of Fe-LG had adverse effects on the growth performance by decreasing BW of broilers on day 21 and ADG on days 1-21, while increasing FCR in the starter phase. The organ indices of the heart, liver, spleen, and kidneys were reduced in broilers supplemented with 800 mg Fe/kg. Therefore, diet supplemented with 800 mg Fe/kg in the form of Fe-LG had adverse effects on growth performance during the starter phase and organ development in broilers.” in Line 381-386.

Reviewer 2 Report
Reviewers' Comments on a manuscript titled “High dietary organic iron decreased growth performance and induced oxidative stress of broilers”.
Dear Authors
The hypothesis of the study is not clear. For example, why 800mg/kg of Fe-LG was used when the current recommendation is around 80mg/kg of supplemental Fe for poultry, and the literature suggests that levels above 590mg/kg of Fe in diet have toxic effects? What was the rationale behind using an exceptionally high level of Fe-LG? Must be clarified in the manuscript. If the purpose was solely to induce oxidative stress, this does not mimic commercial husbandry. Birds on a commercial farm will never experience this stress as they will never be exposed to such a high level of dietary Fe levels in poultry. Please clarify what the take-home message from this piece of research is?
The whole manuscript is full of grammatical errors. Please check and correct.
Few comments to consider that may allow improving the manuscript quality:
Line 3: It should be “in” broilers.
Line 11: expose to should be “exposure to”
Line 12: organic Fe “have” should be “has”
Line 13: “evaluated” should be “evaluate”
Line 13: what do you mind by “kind of”organic Fe? Not clear? Is it organic or inorganic?
Line 63: What is the level of Fe to be considered an overdose. Check reference and include this info.
Line 71: the word should be “toxic” not toxicity”
Line 89: The numbers don’t add up. If you have 960 birds in total. 5 groups each having eight replicates, then it should be 24 birds/replicate. Please check. If only 20/pen were used, then the total number of birds should be 800.
Table 1: Clarify “Flour” of which cereal?
Table 1: Unit of energy should be either MJ/kg or Kcal/kg.
Table 1: Was it the total AA of SID AA.
Line 92: Was Fe-LG added on the top of the formulation, or was it part of the feed formulation? Must mention details.
Line 93: Was feed mash or pelleted? Add info.
Line 131: The duodenum samples: were they tissue samples or digesta? Not clear
Line 140-148: Instead of a paragraph, add a table with three columns, i.e. Type of analysis, the kit used and reference of the methodology used.
Line 150 and 152: The information should be provided in the Table. Give the reference for each kit used.
Table 2: Add another column at the end to add a reference for each gene.
Page 7: Is there any biological significance of differences observed in organ index? Any reference?
For blood biochemical parameters (Table 6). Do you have normal ranges for each item measured? If yes, add those ranges to the Table. Is increase or decrease out of range or within range.
Discussion section:
Line 258: Fe overload: What is the dosage that can initiate Fe overload (check reference 11 and provide this information).
No one supplements such high doses of supplemental Fe in poultry diets, so would it be correct that Fe source will not cause oxidative stress under farm conditions?
Line 276: what kind of changes of lipid: not clear.
Line 311: Under which circumstances would you expect birds to be exposed to 800mg Fe iron in the chicken diet.?
Line 323: give reference and mention forms of iron.
Line 325: what do you mean by “kind of”. Expression is not appropriate.
Conclusion:
Line 358-359: This is not true. Only juvenile birds showed an adverse effect. No adverse effects were noted in growing /adult birds. The sentence needs to be amended.
Reviewer 3 Report
This is a very interesting study, which brings further insights to organic Fe supply in the diet of broilers. The manuscript is very clear and very well written. The discussion could be further improved using shorter sentences, in order to increase clarity. The only issue I would suggest to improve is to stress the importance of the Fe sources used, as it is the core novelty of this study. Several studies that were previously done have reported most of the biochemical, histological and transcriptomic changes that were observed in this study but, to my knowledge, this is one of the first studies focusing on chelated forms of Fe. So I would suggest highlighting the value of this research by improving the discussion of the results while taking into consideration the bioavailability of Fe sources used in this study.
Specific comments are reported in the attached file.

Round 2
Reviewer 2 Report
Dear Authors,
I understand that it's a tolerance plus efficacy study, but I don't understand why the focus is mainly on the highest supplemental dose of organic Fe. As a reader, I am interested in knowing the recommended dosage of Fe-LG? Should I use 40 or 80ppm? What is the use level of the Fe-LG that you are recommending. You have made the whole discussion about the adverse effects of the Fe-LG at 800ppm. Your abstract, discussion and conclusion sections need revision to include the use level. Otherwise, as a poultry producer, nutritionist and researcher, the manuscript is of no interest to the reader as such a high level of Fe source is never intended for a broiler diet. Without even your research trial, I already know that level of Fe above 500ppm has adverse effects on broilers. I would like to read your manuscript because you are exploring an organic Fe chelate, and I would like to know the recommended level.
The manuscript has been improved, but the message conveyed has nothing we diddn't knew before. The manuscript should be about the use of Fe-LG in the broiler diet. The manuscript should highlight your recommended level and tell the reader about the safety margin.
The manuscript needs further revision.
